

# South Atlantic Anomaly during ascending and maximum phase of solar cycle 24

Khairul Afifi Nasuddin[1], Mardina Abdullah[1,2], Nurul Shazana Abdul Hamid[3]

[1]Department of Electrical, Electronic and Systems Engineering, Universiti Kebangsaan Malaysia, Bangi, 43600, Malaysia
[2]Space Science Center (ANGKASA), Institute of Climate Change, Universiti Kebangsaan Malaysia, Bangi, 43600, Malaysia
[3]Department of Applied Physics, Faculty of Science and Technology, Universiti Kebangsaan Malaysia, Bangi, 43600, Malaysia

*Correspondence to*: Nurul Shazana Abdul Hamid (shazana.ukm@gmail.com)

**Abstract.** For this research, four regions have been studied which are the South Atlantic Anomaly (SAA) region, low latitude region, middle latitude region and high latitude region. The active period which is the period when the geomagnetic storm occur chosen to be analyzed is on 6 August 2011 and 12 April 2014 and the normal period, a period when no geomagnetic storm happen is on 24 July 2011 and 14 May 2014. Year 2011 is chosen to be analyze in order to study the SAA region during the ascending phase of the solar cycle 24 and in year 2014, where the occurrence of the maximum phase of solar cycle 24
occur. The research is carried since there is no clear characterization of the SAA during ascending as well as maximum phase based on power spectrum analysis method. The Earth's magnetic field component chosen to be analyzed is the horizontal intensity (H) due to its sensitiveness regarding geomagnetic activeness. From the research conducted, the result reveal SAA region has a tendency to be persistent during both period compare to other region during both phases. Other regions in the research experience a tendency to be persistent and antipersistent.

**1 Introduction**

The appearance of the sky is dependent on the sun's altitude (Bahali et al., 2018). Sun-oriented winds with high densities and strong magnetic fields (Umar et al., 2019). The sun produces solar flare which is related to the geomagnetic activity. Together with coronal mass ejections, a flare is an explosive event that releases high energy protons and electrons, including intense radiation in all wavelengths and can affect the Earth's atmosphere (Bahari et al., 2011).

Understanding the coupling mechanisms between various processes and phenomena in the solar–terrestrial system remains a considerable challenge (Hall, 2014). Earth satellite communication and other applications of satellite technology have made the study of space environment, including the ionosphere, more popular in recent times (Rabiu et al., 2017). It aims to obtain and identify the principle, concept and laws that include natural phenomena (Majid et al., 2015). The ionosphere is a region of the upper atmosphere which is important for the propagation of High-frequency (HF) radio communication


(Bello et al., 2017). Study of ionospheric disturbances is of interest because it is not only a scientific task but also an important applied problem (Blagoveshchensky and Sergeeva, 2019).

One of ionospheric disturbances occur originating from the sun is the geomagnetic storm. These storms have been known to give rise to much devastating effect on electric grid and other technologies (Okpala and Ogbonna, 2018). Geomagnetic storms are one of the most striking phenomena in the magnetic field that we can measure at the Earth's surface
(Papa and Sosman, 2008). This research is conducted in order to study the characterization of the South Atlantic Anomaly (SAA) by comparing with different regions during the occurrence of geomagnetic storm and during normal period with the analyzation of different phase of solar cycle.

The SAA region can be describe as a low intensity magnetic field region. Study has been made by Domingos (2017) on the impact of SAA on the space weather. One of the dangers to spacecraft in orbit around the Earth is the presence of large
doses of radiation encountered in space (Schaefer et al., 2016).

## 1.1 The South Atlantic Anomaly

SAA is formed because of the noncoincidence of the geomagnetic dipole axis and the Earth's rotating axis (Zou et al., 2015).The asymmetry of the geomagnetic field about the Earth's geomagnetic center results in an enhanced interaction of the radiation belt particles with the Earth's atmosphere in the regions of low field strength (Jr and Gonzalez, 1989). The reduced
magnetic field results in a higher level of high energy particle precipitation in this region than anywhere else on the Earth during solar storms (Cilliers et al., 2009). Such a high radiation level impacts the behavior of on-board oscillators: a rapid change in the frequency can be observed when the satellites fly across this area (Jalabert and Mercier, 2018). In this region spacecrafts get the greatest radiation dose impact which is associated with intensive fluxes of charged particles (protons and electrons) (Grigoryan et al., 2008).

SAA radiation also poses a potential threat to personal and biological systems during low Earth orbit (Olson and Amit, 2006). The hazard of this region is little recognized outside of segments of the space science community, but nearly all spacecraft crossing this area at altitudes of 100 km to 1000 km have been damaged or degraded in performance to some extent (Heirtzler et al., 2002). Satellites in low-Earth orbit pass though the SAA periodically, exposing them to several minutes of strong radiation each time: the International Space Station requires extra shielding to deal with this problem and astronauts on
extravehicular activity try to avoid it (Anderson et al., 2018). In this region the intensity of the geomagnetic field is a global minimum and therefore mirror points of the trapped charged particles get to lower altitudes and in interacting with the denser atmosphere they precipitate causing ionization much more intensely in this anomaly region than at other longitudes (Trivedi et al., 2005). The trapped radiation environment, consisting of large amounts of energetic charged particles, can be potentially harmful to human beings and space vehicles immerged in it (Qin et al., 2014). This high radiation zone at a low altitude has
potential threat to low orbit spacecrafts, such as weather-forecast satellites and manned missions (Ye et al., 2017).Although most of the effects of SAA have been studied for its external interactions with the upper atmosphere and magnetosphere, the





way SAA changes in time, space and magnitude can provide important clues about our planet's interior and dynamics (Santis and Qamili, 2010).

The SAA is of great interest to the scientific community as well as to the public because high-intensity solar corpuscular radiation can penetrate the Earth's orbit in the SAA region due to the weak main magnetic field (Koch and Kuvshinov, 2015). Knowledge of the radiation fields specific to the SAA is of great importance for the radiation safety of cosmonauts during a long orbital flight (Benghin et al., 1992). In order to analyze the characteristic of the SAA, comparison of the SAA with different region is made. The method apply is power spectrum analysis and Hurst exponent and the period taken to be analyze is during the occurrence of geomagnetic storm (active period) and during no occurrence of geomagnetic

storm (normal period). Different phase of solar cycle 24 is studied in order to gain more knowledge on the characteristic of SAA.

**1.2 Review of SAA and Power Spectrum Analysis**

Research on the SAA with applying power spectrum analysis method has been conducted by Nasuddin et. al (2019). In this research, the SAA is studied by comparing with regions outside the SAA and the Hurst exponent is obtained. Based on

the research conduct on the analyzation of SAA during active period on 11 March 2011 and normal period 3 February 2011, the outcome reveal SAA has a tendency to be persistent during active period (11 March 2011) and normal period (3 February 2011) when compare to other region such as middle latitude region and high latitude region in the research. As an extension to do work done by Nasuddin et al. (2019), four regions which are the SAA region, low latitude region, middle latitude region and high latitude region are being studied during different year to examine the influence of solar cycle 24 on the regions.

Therefore, a research carried out regarding SAA can be used as a reference in the satellite launch and increase knowledge in the field of geomagnetic field (Nasuddin et al., 2015).

The research on the SAA throughout the solar cycle has been made by Domingos et al. (2017). Evolution of the SAA particle flux can be seen as the result of two main effects, the secular variation of the Earth's magnetic field and the modulation of the protons density of the inner Van Allen radiation belt during the solar cycle (Domingos et al., 2017). In order to learn on

the evolution of the particle flux anomaly, the method Principal Component Analysis (PCA) of either Polar Orbiting Environmental Satellites particle flux or Cloud-Aerosol Lidar with Orthogonal Polarization dark noise has been applied. The progress made allow to estimate the influence of the SAA on the space weather.

The disturbance storm time index (Dst) is a measure of geomagnetic activity used to assess the magnetic storms and it is affected by solar output (Nigam et al., 2017). For the study on the effect of coronal mass ejection on Earth's magnetic

field during ascending phase of solar cycles 23-24, it has been conducted by Nigam et al. (2017). In the research, the studied of the width as well as the speed of the coronal mass ejections along with the Dst index has been made throughout the ascending phase of solar cycles 23 as well as 24. Based on the studied, the outcomes indicate the correlation between Dst and coronal mass ejections on behalf of the ascending phase of solar cycle 24 is less in comparison to the ascending phase of solar cycle 23.





The nature of the ionosphere during solar minimum is important to be study in order to understand the respond of ionosphere toward changes due to Earth's magnetic field and solar activity (Bahari et al., 2017). Bahari et al. (2017) conduct a research on the variation of total electron content during solar minimum. The intention is to examine the variation of the total electron content concerning Malaysia on geomagnetic activity, diurnal as well as seasonal solar by applying Global Positioning System. Based on the research, the solar variation had more influence on the variation of total electron content in comparison
to geomagnetic variation.

Experimentation on the Hurst exponents of the geomagnetic horizontal component during quiet and active periods has been performed by Hamid et al. (2009). In this experiment, the aim is to describe the fractal properties represented by the Hurst exponent by analyzing the geomagnetic horizontal component, H during active day and normal day at stations in the Cebu and Davao in the Philippines. Based on the research, the fractal parameter can be apply to characterize the variation of
geomagnetic horizontal component, H during active and quiet period. Further studies need to be carried out on data from other stations and also for longer periods of time (Hamid et al., 2009).

The main effect of the interaction between the interplanetary magnetic field, geomagnetic field and the solar wind is that the geomagnetic field is compressed on the sunward side, giving rise to a diurnal variation at mid latitudes (Hamid et al., 2010). In the research conducted by Hamid et al. (2010), the investigation and study on the scaling and fractal properties of
the horizontal component of the geomagnetic field is conducted. The data in this research is obtain from Langkawi, Malaysia and from Davao, Philippines and power spectrum technique, rescaled range analysis and detrended fluctuation analysis is apply. The data set apply in this research is detailed for one month period of quiet geomagnetic activity verified at 1 sample/minute. In this study, it is identify that the fractal methods can be apply to quantify as well as characterize the horizontal geomagnetic field component.

The studies on Monofractal and multifractal characterization of geoelectrical signals measured in southern Italy has been conducted by Telesca et al. (2003). In this research, a period from January 2001 to February 2002 with four stations set up in Basilicata region (southern Italy) have been apply fractal tools in order to characterize the temporal variations in the dynamics of hourly geoelectrical signals. Detrended fluctuation analysis, Higuchi method and Lomb Periodogram are Monofractal method apply in order to identify a large range of timescales scaling behaviour. The multifractal formalism
indicates to the detection of a set of parameters, originating from the multifractal spectrum shape as well as determining the 'complexity' of the signals. In the study of seemingly complex phenomena, like those generating geoelectrical signals, methodologies able to determine time scale structures in observational time series are particularly useful tools to obtain information on the features and on the causes of variation at the different time scales (Telesca et al., 2003).

## 2 Methodology

In the methodology, the power spectrum analysis and Hurst exponent is being studied in order to learn the characteristic of a region. The period of the research focus during the ascending as well as maximum phase of solar cycle 24.





The station involve are stations in the SAA region as well as stations in other region such as low latitude region, middle latitude region and high latitude region.

## 2.1 Stations and locations

130        The station name as well as the IAGA code, the geodetic latitude and geodetic longitude is display in Table 1 till Table 4 and Fig.1  represent the stations position in year 2011, the ascending phase of solar cycle 24 as well as year 2014, the maximum phase of solar cycle 24 occur.

Four region are analyze which are the SAA region, located at $0^0$ to $50^0$ S and from $90^0$ W to $40^0$ E. The low latitude region placed at $0^0$ to $30^0$ latitude. The middle latitude region position at $30^0$ to $60^0$ latitude. For the high latitude region, it is

situated at $60^0$ to $90^0$ latitude.

**Table 1**. Stations in the SAA region.

| Station name | IAGA Code | Geodetic latitude | Geodetic longitude |
| --- | --- | --- | --- |
| Hartebeesthoek | HBK | -25.883 | 27.707 |
| Hermanus | HER | -34.425 | 19.225 |
| Keetmanshoop | KMH | -26.541 | 18.110 |
| Tsumeb | TSU | -19.202 | 17.584 |
| Huancayo | HUA | -12.038 | 284.682 |
| Ascension Island | ASC | -7.949 | 345.624 |

**Table 2.** Stations in the low latitude region.

| Station name | IAGA Code | Geodetic latitude | Geodetic longitude |
| --- | --- | --- | --- |
| Guimar-Tenerife | GUI | 28.321 | 343.559 |
| Addis Ababa | AAE | 9.035 | 38.766 |
| Kourou | KOU | 5.210 | 307.269 |
| Mbour | MBO | 14.392 | 343.042 |
| San Juan | SJG | 18.111 | 293.850 |
| Tamanrasset | TAM | 22.792 | 5.530 |

140        **Table 3.** Stations in the middle latitude region.

| Station name | IAGA Code | Geodetic latitude | Geodetic latitude |
| --- | --- | --- | --- |
| Lviv | LLV | 49.900 | 23.750 |
| Hel | HLP | 54.608 | 18.817 |
| Niemegk | NGK | 52.072 | 12.675 |
| Nagycenk | NCK | 47.630 | 16.720 |
| Uppsala | UPS | 59.903 | 17.353 |


**Table 4.** Stations in the high latitude region

| Station name | IAGA Code | Geodetic latitude | Geodetic longitude |
|---|---|---|---|
| Abisko | ABK | 68.358 | 18.823 |
| Hornsund | HRN | 77.00 | 15.550 |
| Iqaluit | IQA | 63.757 | 291.489 |
| Sodankyla | SOD | 67.369 | 26.630 |
| Lycksele | LYC | 64.612 | 18.748 |

The position of the stations are display in Fig.1. For the SAA region, its station is represent with the black circle. For

the low latitude region, it is represent with the cyan circle. For the middle latitude region, the stations are symbolize with red

circle and for the high latitude region, it is denote with the magenta circle. From Fig. 1, the selection of stations are based in

the area of the dotted line which is in the range of the -90 to 40 degrees of longitude. For the trapezium shape, it represent the

SAA region.

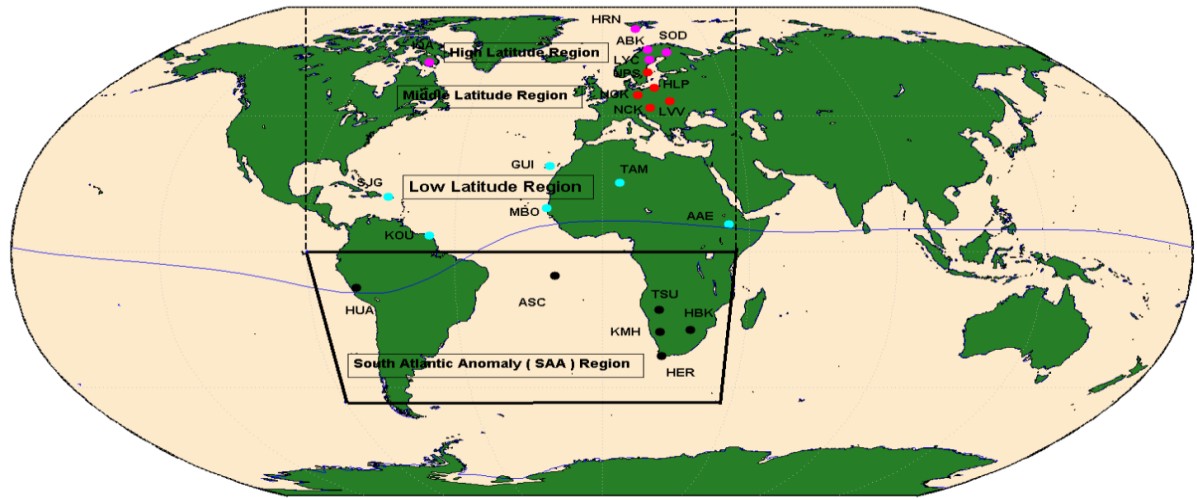

**Figure 1: The stations involve in the research during ascending and maximum phase of solar cycle 24.**

**2.2 Power Spectrum Analysis**

Power spectrum implies the power of each frequency component of the source time domain waveform. In relation to

Fourier analysis, any physical signal can be able to be decomposed into a number of discrete frequencies, or a spectrum of

frequencies over a continuous range. On behalf of the power-spectral density function, which is characterized as $S_m$, can be

designated as,

$$S_m = \lim_{N \to \infty} \left[ 2|Y_m|^2 \big/ N\delta \right], m = 1, 2, 3, \ldots, \frac{N}{2}, \tag{1}$$

Meant for representing a discrete time series, denoted by $y_n$, n = 1,2,3…….,N, whereby δ is the time amid in succession

*n.*

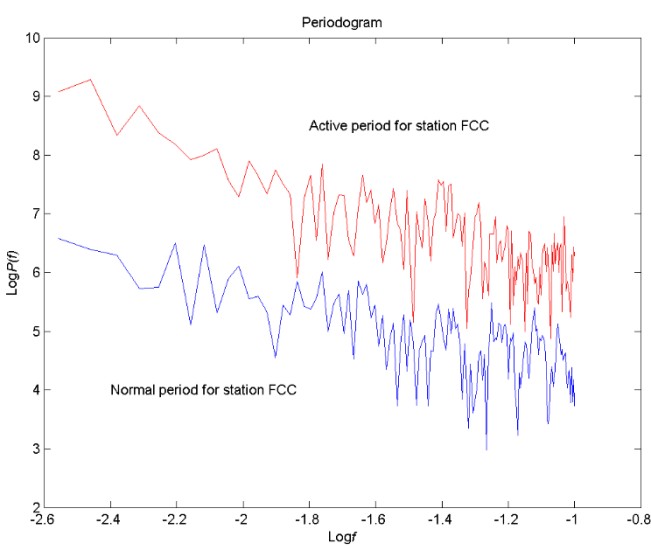

**Figure 2: An example of periodogram for active period and normal period for station Fort Churchill (FCC).**

Power spectrum of a time series can be illustrate as distribution of power into frequency element constituting of that signal. An example of power spectrum, p(f) versus frequency, f in log-log scale is describe in Fig. 2.

Figure 2 represent the periodogram for active period characterized by the red colour and the normal period characterized by the blue colour for station FCC. In other cases, for a self-affine time series, the power-spectral density, denoted

as $S_m$ is depicted comprising a power-law dependence on frequency

$$S_m \sim f_m^{-\beta}, m = 1,2,3,........,\frac{N}{2},$$   (2)

It is to be indicated that $f_m = m/N$ whereby the importance of spectral exponent, $\beta$ is to determine the strength of persistence in a time series.

### 2.3 Hurst exponent

Figure 3 display the periodogram for station FCC on 10 October 2011. From the periodogram, the value of spectral exponent, $\beta$ is obtained from the negative slope of the straight line plot p (f) versus f in log-log scale. From the spectral exponent, $\beta$, the Hurst exponent can be achieved. The spectral exponent, $\beta$ is apply in $H_{PS} = (\beta - 1)/2$ when $1 < \beta < 3$ and $H_{PS} = (\beta + 1)/2$ for $-1 \le \beta < 1$.

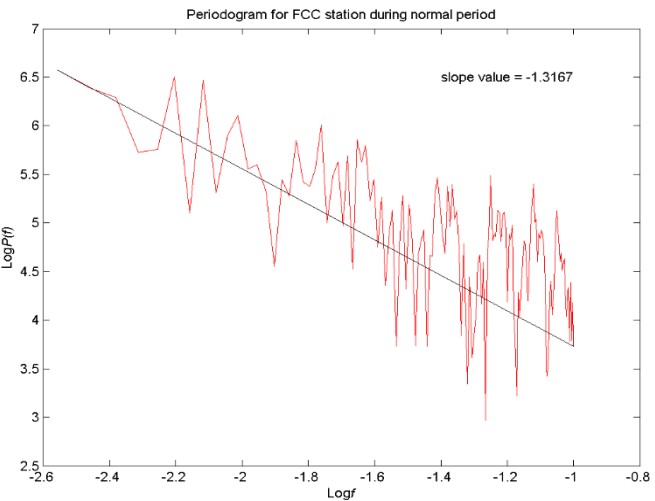

Figure 3: The periodogram for station FCC on 10 October 2011.

## 2.4 Persistent, Antipersistent and Random Noise

From the Hurst exponent, $H_{PS}$ the characteristic of a region can be determine. Time series with $0 < H_{PS} < 0.5$ are identified as antipersistent. For time series with $0.5 < H_{PS} < 1$ are described as persistent. For $H_{PS} = 0.5$ means a random series. Table 5 explain the characteristic of persistent, antipersistent and random series.

**Table 5.** Characteristic of persistent, antipersistent and random series

| Hurst exponent value | Characteristic |
| --- | --- |
| Persistent | If the Hurst exponent is within the range of $0.5 < H_{PS} < 1$, it can be taken as both that another high value is likely to be followed by another high value in the series and that the values will be high for a long time to come. |
| Antipersistent | For a Hurst exponent in the range of $0 < H_{PS} < 0.5$ indicates a time series with long-term switching between high and low values in adjacent pairs, implying a single high value may be followed by a low value and the succeeding value will tend to be high, with this tendency changing between high and low values and remaining for a long time into the hereafter. |
| Random noise | For $H_{PS} = 0.5$ signify a random series. It can also mean data are not correlated – that is, no dependence between current and past data (Nasuddin et al., 2019). |





### 2.5 Period of study

In the period of study, year 2011 and year 2014 have been analyze. Year 2011 is analyze since the occurrence of the
ascending phase of solar cycle 24 happen and year 2014 is when the maximum phase of solar cycle 24 occur. The active period in this studies is the period where the geomagnetic storm take place. During the active period for this research, the Dst index value is consistently below -30 nT. The normal period is when no geomagnetic storm happen.

Figure 4 indicate the Dst index and Kp index for the active period, 6 August 2011 and normal period, 24 July 2011. During the active period, 6 August 2011, a moderate storm occur since the Dst index reach a value in the range between -50
nT to – 100 nT. It occur several times starting from 1 UT until 2 UT and continue from 5 UT until 23 UT. On the same day, 6 August 2011, an intense storm happen since the DST index value reach -104 nT and -115 nT on 3 UT and 4 UT. A weak storm take place as the DST Index drop to -47 nT on 24 UT. The Kp index reveal a moderate geomagnetic storm happen with Kp index, 6 and a minor geomagnetic storm with Kp index, 5 on 6 August 2011. On 24 July 2011, no geomagnetic storm occur with Kp Index on that period doesn't reach 5 and exceed 5, indicating no geomagnetic storm event.

The red line in the Dst Index indicate the threshold for a geomagnetic storm to happen. The existence of geomagnetic storm occur when the Dst Index is under -30 nT. On 12 April 2014, the Dst Index reveal a moderate storm from 1 UT to 23 UT. It is follow by a weak storm on 24 UT as it reach -49 nT. The Kp index reading indicate a minor geomagnetic storm occurrence with Kp index displaying 5. On the normal period, 14 May 2014, no geomagnetic storm occur with the Dst Index reading above -30 nT and the Kp Index showing a value of 1 and 2.

There are a number of geomagnetic storm occur during year 2011 and 2014. The selection of the active period, 6 August 2011 is due to several types of geomagnetic storm occur during that period. It is observe that a weak, moderate and intense storm happen on that day. For the active period, 12 April 2014, it is chosen since the maximum phase of solar cycle 24 occur on April 2014.

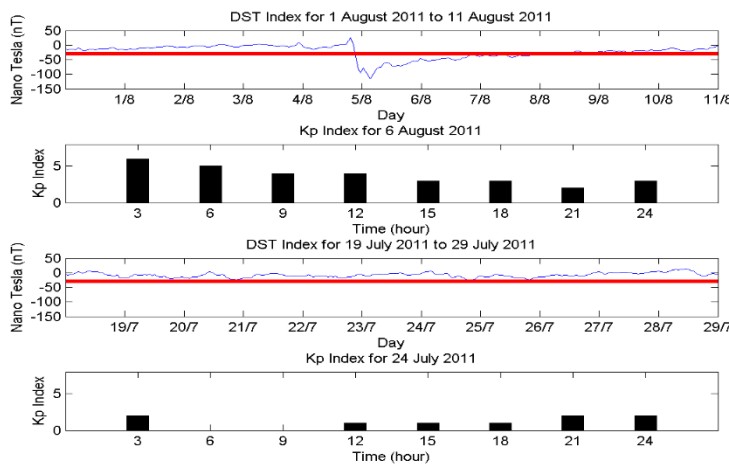

**Figure 4: Dst Index and Kp Index for active period (6 August 2011) and normal period (24 July 2011).**

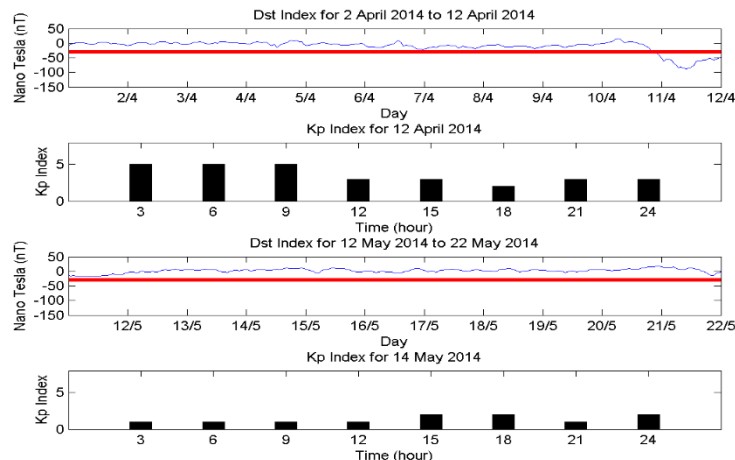

**Figure 5: Dst Index and Kp Index for active period (12 April 2014) and normal period (14 May 2014).**

### 2.6 Component of the Earth's magnetic field to be analyze

The component of the Earth's magnetic field can be analyze in different ways. The F component is the strength of the Earth's magnetic field. The D and Z component of the Earth magnetic field can be apply in studying the earthquake. The method to study the earthquake is polarization ratio. The polarization ratio analysis method has successfully detected an anomaly preceding the main earthquake (Yusof et al., 2019). The H-component or Horizontal intensity is suitable to study the geomagnetic storm.



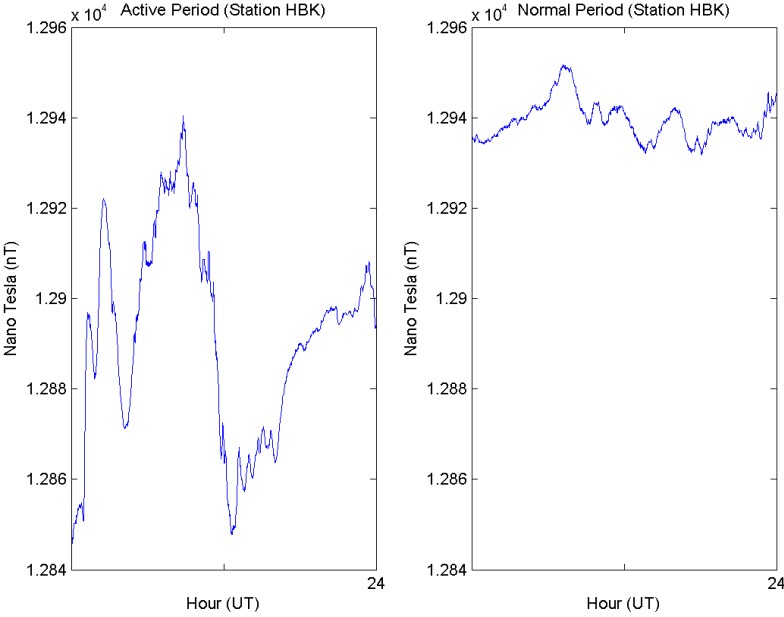

**Figure 6: Comparison between active period and normal period for station HBK.**

The difference between active period and normal period can be seen on Fig. 6. It indicate the sensitiveness of the Horizontal intensity during geomagnetic storm represent by station HBK on 6 August 2011 (active period) and the Horizontal intensity characteristic during 24 July 2011 (normal period). For a  magnetic storm, the start of a magnetic storm which is often

portrayed by a global sudden rise in the Horizontal intensity mentioned as the storm sudden commencement or SSC. Following the storm sudden commencement, the Horizontal intensity usually remains on top of its average level for several hours whereby this phase is known as the initial phase of the storm. Subsequently, a great global decrease in the Horizontal intensity begin, implying the forming of the main phase of the storm. The research conducted compare the period during geomagnetic storm occurrence and normal period whereby the Horizontal intensity is chosen based on its sensitiveness toward geomagnetic

activity level.

## 3 Result and Discussion

In the result and discussion, four region are analyzed. The analyzation is made in order to understand the characteristic of the SAA region during ascending and maximum phase of solar cycle 24. Table 6 until Table 13 reveal the results for the study. Table 6, Table 8, Table 10 and Table 12 indicate the result during active period (6 August 2011) and normal period (24

July 2011) for SAA region, low latitude region, middle latitude region and high latitude region while Table 7, Table 9, Table 11 and Table 13 display the result for active period (12 April 2014) and normal period (14 May 2014) for SAA region, low latitude region, middle latitude region and high latitude region.





**Table 6.** Result for active period (6 August 2011) and normal period (24 July 2011) for stations in the SAA region.

| Station | Hurst exponent on 6 August 2011 | Character during active period (6 August 2011) | Mean of the Earth Magnetic Field Strength (nT) | Hurst exponent on 24 July 2011 | Character during normal period (24 July 2011) | Mean of the Earth Magnetic Field Strength (nT) |
|---|---|---|---|---|---|---|
| HBK | 0.8110 ± 0.0674 | Persistent | 28350 | 0.5570 ± 0.0674 | Persistent | 28360 |
| HER | 0.8231 ± 0.0629 | Persistent | 25750 | 0.5609 ± 0.0661 | Persistent | 25760 |
| KMH | 0.7804 ± 0.0752 | Persistent | 27210 | 0.5716 ± 0.0685 | Persistent | 27230 |
| ASC | 0.6247 ± 0.0430 | Persistent | 28330 | 0.5580 ± 0.0544 | Persistent | 28370 |
| HUA | 0.7189 ± 0.0704 | Persistent | 25230 | 0.6129 ± 0.0670 | Persistent | 25310 |

**Table 7.** Result for active period (12 April 2014) and normal period (14 May 2014) for stations in the SAA region.

| Station | Hurst exponent on 12 April 2014 | Character during active period (12 April 2014) | Mean of the Earth Magnetic Field Strength (nT) | Hurst exponent on 14 May 2014 | Character during normal period (14 May 2014) | Mean of the Earth Magnetic Field Strength (nT) |
|---|---|---|---|---|---|---|
| HBK | 0.6719 ± 0.0551 | Persistent | 28320 | 0.5661 ± 0.0719 | Persistent | 28330 |
| HER | 0.6587 ± 0.0543 | Persistent | 25630 | 0.5520 ± 0.0734 | Persistent | 25640 |
| KMH | 0.6654 ± 0.0569 | Persistent | 27080 | 0.6135 ± 0.0614 | Persistent | 27090 |
| TSU | 0.6957 ± 0.0535 | Persistent | 29510 | 0.6128 ± 0.0679 | Persistent | 29520 |
| HUA | 0.7167 ± 0.0638 | Persistent | 25040 | 0.6829 ± 0.0739 | Persistent | 25100 |

In this study, the analysis on SAA region during solar cycle 24 is made. Year 2011 is the ascending phase of solar cycle 24 and in year 2014, the maximum phase of solar cycle 24 occur. SAA region is situated at an altitude of 200-800 km above the Earth's surface and it experience a low Earth's magnetic field strength compare to other region. During the ascending

phase of solar cycle 24, 5 stations has been analyze during the active period, 6 August 2011 and normal period, 24 July 2011. Thus 5 station analyze during the active period, 6 August 2011 which are station HBK, HER, KMH, ASC and HUA exhibit a persistent characteristics. The same situation experience during normal period, 24 July 2011 whereby station HBK, HER, KMH, ASC and HUA present a persistent features. The mean of the Earth's magnetic field strength experience by stations in the SAA region during active period, 6 August 2011 and normal period 24 July 2011 is low with station HBK during active

period, 6 August 2011 and normal period, 24 July 2011 reveal a mean of the Earth's magnetic field strength of 28350 nT and 28360 nT. It is the highest mean of the Earth's magnetic field strength SAA region experience in this period while the lowest experience by station HUA with 25230 nT during active period (6 August 2011) and 25310 nT during normal period (24 July 2011).





The study continue in year 2014, in this year the solar cycle exhibit a maximum phase solar cycle 24. Period of study
is on 12 April 2014 (active period) and 14 May 2014 (normal period). Stations in the SAA region during this period a tendency
to be persistent. Station HBK, HER, KMH, TSU and HUA during 12 April 2014 (active period) reveal a persistent
characteristics. It can be seen the mean of the Earth's magnetic field strength experience is not large with station HBK (28320
nT), HER (25630 nT), KMH (27080 nT), TSU (29510 nT) and HUA (25040 nT).

During normal period, 14 May 2014, the same characteristic is experience by stations in the SAA region. The Hurst
exponent value for station HBK, HER, KMH, TSU and HUA is experiencing a persistent characteristics. The mean of the
Earth's magnetic field indicate a low Earth magnetic field strength with station HBK (28330 nT), HER (25640 nT), KMH
(27090 nT), TSU (29520 nT) and HUA (25100 nT).

Based on observation, on year 2011, during the ascending phase solar cycle 24 and on year 2014 where the maximum
phase solar cycle 24 happen, SAA region experience a tendency to persistent. It can also be seen the tendency to be persistent
of SAA region during active period and normal period. Analysis is made and it is observe that the SAA region experience a
low Earth's magnetic field compare to other region and this may be a factor that contribute to SAA region to experience this
characteristic.

**Table 8.** Result for active period (6 August 2011) and normal period (24 July 2011) for stations in the low latitude region.

| Station | Hurst exponent on 6 August 2011 | Character during active period (6 August 2011) | Mean of the Earth Magnetic Field Strength (nT) | Hurst exponent on 24 July 2011 | Character during normal period (24 July 2011) | Mean of the Earth Magnetic Field Strength (nT) |
|---|---|---|---|---|---|---|
| AAE | 0.4162 ± 0.0674 | Antipersistent | 36220 | 0.4474 ± 0.0654 | Antipersistent | 36290 |
| TAM | 0.7745 ± 0.0779 | Persistent | 37680 | 0.4618 ± 0.0702 | Antipersistent | 37720 |
| MBO | 0.6455 ± 0.0599 | Persistent | 32640 | 0.6084 ± 0.0717 | Persistent | 32700 |
| SJG | 0.5891 ± 0.0497 | Persistent | 37580 | 0.5385 ± 0.0712 | Persistent | 37620 |
| KOU | 0.6355 ± 0.0526 | Persistent | 29120 | 0.5374 ± 0.0516 | Persistent | 29180 |

**Table 9.** Result for active period (12 April 2014) and normal period (14 May 2014) for stations in the low latitude region.

| Station | Hurst exponent on 12 April 2014 | Character during active period (12 April 2014) | Mean of the Earth Magnetic Field Strength (nT) | Hurst exponent on 14 May 2014 | Character during normal period (14 May 2014) | Mean of the Earth Magnetic Field Strength (nT) |
|---|---|---|---|---|---|---|
| GUI | 0.6635 ± 0.0505 | Persistent | 35890 | 0.5325 ± 0.0652 | Persistent | 35930 |
| KOU | 0.6231 ± 0.0405 | Persistent | 28950 | 0.6229 ± 0.0781 | Persistent | 29010 |
| MBO | 0.6302 ± 0.0578 | Persistent | 32690 | 0.6897 ± 0.0758 | Persistent | 32750 |
| SJG | 0.6456 ± 0.0575 | Persistent | 37280 | 0.5776 ± 0.0623 | Persistent | 37300 |
| TAM | 0.6438 ± 0.0567 | Persistent | 37740 | 0.4658 ± 0.0638 | Antipersistent | 37800 |





On 6 August 2011 (active period) and 24 July 2011 (normal period), stations in the low latitude region experience a characteristic different compare to the SAA region. Stations in the low latitude region has a mixture of persistent as well as antipersistent characteristics. This can be observe on 6 August 2011 (active period) where station AAE exhibit an antipersistent feature. Station TAM, MBO, SJG and KOU reveal a persistent value. It can be refer to Table 8. It is observe the mean of the

Earth's magnetic field strength in low latitude regions is higher compare to SAA region during 6 August 2011 (active period) and 24 July 2011 (normal period) with station AAE (36220 nT), TAM (37680 nT), MBO (32640 nT), SJG (37580 nT) and KOU (29120 nT).

On 24 July 2011 (normal period), as observe the increase of the mean of the Earth's magnetic field strength, show the tendency of stations in the low latitude regions to experience a difference feature compare to the SAA region, with station

AAE exhibit a Hurst exponent value of $0.4474 \pm 0.0654$ meaning an antipersistent value. Station TAM reveal a Hurst exponent of $0.4618 \pm 0.0702$ indicating an antipersistent characteristics. During this period, station AAE experience an Earth's magnetic field value of 36290 nT and station TAM with 37720 nT. It can be seen the mean of the Earth's magnetic field experience is larger compare to the SAA region.

Other stations in the low latitude region during the normal period dated on 24 July 2011 display a persistent value

with station MBO revealing a mean of the Earth's magnetic field strength of 32700 nT, station SJG showing a mean fo the Earth's magnetic field strength of 37620 nT and station KOU with 29180 nT as of the mean of the Earth's magnetic field strength it experience.

Research has been made and during April 2014, it is where the maximum phase of solar cycle 24 occur. We can see the influence of solar cycle in year 2014 as compared to year 2011 in the low latitude regions. In the active period (6 August

2011) and normal period (24 July 2011), stations in the low latitude regions exhibit a mixture of persistent and antipersistent characteristics.

During maximum phase of solar cycle, large number of sunspots appear. The activities occurring in the sun has a significant impact on the Earth. As for the active period (12 April 2014), stations in the low latitude region experience a persistent characteristic as exhibit by station GUI, KOU, MBO, SJG and TAM. It may be due to the influence of the maximum

phase of solar cycle 24 for low latitude region to experience this characteristics since during the ascending phase of solar cycle 24, stations in low latitude region exhibit a mixture of antipersistent and persistent value. The mean of Earth's magnetic field for station GUI ( 35890 nT), KOU (28950 nT), MBO (32690 nT), SJG (37280 nT) and TAM (37740 nT) during active period, 12 April 2014.

In the normal period dated 14 May 2014, stations in the low latitude region exhibit a difference characteristics with

station TAM displaying an antipersistent value with mean of Earth's magnetic field strength of 37800 nT. Other stations such as GUI, KOU, MBO and SJG produce a persistent characteristics. The mean of the Earth's magnetic field strength for station GUI, KOU, MBO and SJG is 35930 nT, 29010 nT, 32750 nT and 37300 nT.

A difference characteristics can be observe between the SAA region and low latitude region. SAA region in this study reveal a tendency to be persistent while low latitude region experience a characteristics of antipersistent as well as persistent.





As the mean of the Earth's magnetic field strength of a region increased, regions tend to experience a more antipersistent as well as persistent characteristics.

**Table 10**. Result for active period (6 August 2011) and normal period (24 July 2011) for stations in the middle latitude region.

| Station | Hurst exponent on 6 August 2011 | Character during active period (6 August 2011) | Mean of the Earth Magnetic Field Strength (nT) | Hurst exponent on 24 July 2011 | Character during normal period (24 July 2011) | Mean of the Earth Magnetic Field Strength (nT) |
|---|---|---|---|---|---|---|
| UPS | 0.5417 ± 0.0483 | Persistent | 51210 | 0.3789 ± 0.0655 | Antipersistent | 51240 |
| LVV | 0.6780 ± 0.0581 | Persistent | 49690 | 0.4527 ± 0.0793 | Antipersistent | 49700 |
| HLP | 0.4359 ± 0.0634 | Antipersistent | 50180 | 0.2806 ± 0.0688 | Antipersistent | 50190 |
| NGK | 0.6057 ± 0.0560 | Persistent | 49200 | 0.4306 ± 0.0586 | Antipersistent | 49210 |
| NCK | 0.7075 ± 0.0573 | Persistent | 48280 | 0.4884 ± 0.0656 | Antipersistent | 48290 |

**Table 11.** Result for active period (12 April 2014) and normal period (14 May 2014) for stations in the middle latitude region.

| Station | Hurst exponent on 12 April 2014 | Character during active period (12 April 2014) | Mean of the Earth Magnetic Field Strength (nT) | Hurst exponent on 14 May 2014 | Character during normal period (14 May 2014) | Mean of the Earth Magnetic Field Strength (nT) |
|---|---|---|---|---|---|---|
| HLP | 0.5041 ± 0.0619 | Persistent | 50280 | 0.5308 ± 0.0686 | Persistent | 50280 |
| LVV | 0.7180 ± 0.0627 | Persistent | 49790 | 0.4441 ± 0.0697 | Antipersistent | 49790 |
| NCK | 0.6693 ± 0.0592 | Persistent | 48380 | 0.5910 ± 0.0680 | Persistent | 48380 |
| NGK | 0.6315 ± 0.0587 | Persistent | 49290 | 0.6295 ± 0.0681 | Persistent | 49290 |
| UPS | 0.4879 ± 0.0761 | Antipersistent | 51310 | 0.7388 ± 0.0663 | Persistent | 51320 |

In year 2011, the active period (6 August 2011) and normal period (24 July 2011) has been studied. The middle latitude region consist of 5 station which are station UPS, LVV, HLP, NGK and NCK. During the active period (6 August

2011), station HLP experience a Hurst exponent of 0.4359 ± 0.0634 meaning an antipersistent value. As for other stations , a persistent value is experience for station UPS, LVV , NGK and NCK. Studied reveal the mean of Earth's magnetic field strength in the middle latitude region is stronger compare to the low latitude region and SAA region with station UPS, LVV, HLP, NGK and NCK exhibit a mean of Earth's magnetic field strength of 51210 nT, 49690 nT, 50180 nT, 49200 nT and 48280 nT.

As of the normal period on 24 July 2011, station UPS, LVV, HLP, NGK and NCK experience an antipersistent characteristics which can be refer to Table 10. The result obtain differ compare to the low latitude region and SAA region and





it may due to the increase of the mean of the Earth's magnetic field strength. The antipersistent value implies a time series with long-term switching between high and low values in adjacent pairs, signifying a single high value may be followed by a low value and the following value will tend to be high, with this tendency to change between high and low values, on-going for a long time into the future.


Solar maximum can be interpret as a regular period of greatest Sun activity during the 11-year solar cycle. In the period of solar maximum, large numbers of sunspots appear. It is to be noted that the increased energy output of solar maxima has the ability to impact the Earth's global climate. In the research on the stations in the middle latitude region on 12 April 2014 (active period), stations in the middle latitude region exhibit a different characteristic in comparing with the stations in low latitude regions and SAA regions. Station UPS present an antipersistent characteristic with Hurst exponent of $0.4879 \pm 0.0761$ with the mean of Earth's magnetic field strength of 51310 nT. Other stations, station NGK, NCK, LVV and HLP reveal a persistent value. The mean of the Earth's magnetic field strength for those stations is 49290 nT for station NGK, 48380 nT for station NCK, 49790 nT for station LVV and 50280 nT for station HLP. In the active period (12 April 2014), the stations in middle latitude region experience a tendency to be persistent as well as antipersistent in contrast to stations in the low latitude region and SAA region. It may be due to influence of the maximum of solar cycle 24 and also the increase of the mean of Earth's magnetic field strength.


During the normal period (14 May 2014), station UPS, NGK, NCK and HLP show a persistent value while station LVV display an antipersistent value of Hurst exponent, $0.4441 \pm 0.0697$ with the mean of the Earth's magnetic field, 49790 nT. Other stations exhibit a mean of the Earth's magnetic field strength of 50280 nT (station HLP), 48380 nT (station NCK), 49290 nT (station NGK) and 51320 nT (station UPS).

Research on the middle latitude region is conducted and observation made reveal middle latitude region experience a characteristic of persistent as well as antipersistent during both period, active period and normal period. The mean of the Earth's magnetic field for middle latitude region is also larger in comparison the low latitude region and SAA region. The studies continue with the high latitude region.


**Table 12**. Result for active period (6 August 2011) and normal period (24 July 2011) for stations in the high latitude region.

| Station | Hurst exponent on 6 August 2011 | Character during active period (6 August 2011) | Mean of the Earth Magnetic Field Strength (nT) | Hurst exponent on 24 July 2011 | Character during normal period (24 July 2011) | Mean of the Earth Magnetic Field Strength (nT) |
|---|---|---|---|---|---|---|
| IQA | $0.2395 \pm 0.0540$ | Antipersistent | 57270 | $0.4431 \pm 0.0715$ | Antipersistent | 57240 |
| SOD | $0.4614 \pm 0.0687$ | Antipersistent | 52730 | $0.3300 \pm 0.0699$ | Antipersistent | 52730 |
| ABK | $0.4574 \pm 0.0818$ | Antipersistent | 53050 | $0.3938 \pm 0.0653$ | Antipersistent | 53040 |
| LYC | $0.3695 \pm 0.0649$ | Antipersistent | 52110 | $0.2371 \pm 0.0711$ | Antipersistent | 52140 |
| HRN | $0.6707 \pm 0.0686$ | Persistent | 54470 | $0.2579 \pm 0.0652$ | Antipersistent | 54440 |





**Table 13.** Result for active period (12 April 2014) and normal period (14 May 2014) for stations in the high latitude region.

| Station | Hurst exponent on 12 April 2014 | Character during active period (12 April 2014) | Mean of the Earth Magnetic Field Strength (nT) | Hurst exponent on 14 May 2014 | Character during normal period (14 May 2014) | Mean of the Earth Magnetic Field Strength (nT) |
|---------|---------|---------|---------|---------|---------|---------|
| ABK | 0.2515 ± 0.0609 | Antipersistent | 53150 | 0.5745 ± 0.0716 | Persistent | 53130 |
| HRN | 0.6831 ± 0.0645 | Persistent | 54520 | 0.6779 ± 0.0594 | Persistent | 54500 |
| IQA | 0.5783 ± 0.0619 | Persistent | 57140 | 0.4412 ± 0.0694 | Antipersistent | 57110 |
| SOD | 0.3659 ± 0.0639 | Antipersistent | 52810 | 0.5937 ± 0.0671 | Persistent | 52820 |
| LYC | 0.1777 ± 0.0628 | Antipersistent | 52210 | 0.6162 ± 0.0746 | Persistent | 52220 |


In year 2011, the ascending phase of the solar cycle 24, research has been made in the high latitude region. The high latitude region during active period (6 August 2011) indicate a majority of antipersistent value which is experience by station IQA, SOD, ABK and LYC. The Hurst exponent value can be refer in Table 12. Station HRN display a persistent value of Hurst exponent, 0.6707 ± 0.0686. Stations in the high latitude region experience a stronger mean of the Earth's magnetic field

strength compare the middle latitude region, low latitude region and SAA region with station IQA (57270 nT), SOD (52730 nT), ABK (53050 nT), LYC (52110 nT) and HRN (54470 nT).

During normal period (24 July 2011), stations in the high latitude region exhibit an antipersistent characteristics with all station IQA, SOD, ABK, LYC and HRN revealing antipersistent value. The mean of the Earth's magnetic field strength is 57240 nT  (station IQA), 52730 nT (station SOD), 53040 nT (station ABK), 52140 nT (station LYC) and 54440 nT (station

HRN). Based on analysis, the increase of the mean of the Earth's magnetic field may contribute to the feature of a regions as the research has been conducted on the middle latitude region, low latitude region and SAA region. In the high latitude region, antipersistent value is more compare to persistent and observations made indicate the increase of the mean of the Earth's magnetic field value.

Year 2011 can be described as the ascending phase of solar cycle 24 while year 2014, the maximum of solar cycle 24

occur. Solar cycle can be interpret as the periodic recurrence of sunspots at Sun's surface. During the event of solar cycle, there occurs periodic change in the solar radiation. The activeness of the Sun is able to produce huge amounts of energetic radiations. In this study dated on 12 April 2014, station ABK, SOD and LYC produce an antipersistent value with mean of Earth's magnetic field strength, station ABK (53150 nT), SOD (52810 nT) and LYC (52210 nT). Station HRN and station IQA reveal a persistent value with mean of Earth's magnetic field strength of 54520 nT for station HRN and 57140 nT for

station IQA which can be refer to Table 13.

On  14 May 2014 (normal period), the high latitude region experience a mixture of antipersistent and persistent characteristic. Station IQA display an antipersistent value of  Hurst exponent, 0.4412 ± 0.0694 with the mean of Earth's magnetic field strength, 57110 nT. Other stations which exhibit a persistent value are station ABK, HRN, SOD and LYC with





the mean of Earth's magnetic field strength, 53130 nT (station ABK), 54500 nT (station HRN), 52820 nT (station SOD) and

52220 nT (station LYC).

In the high latitude region, the mean of the Earth's magnetic field is higher compare to other region in the research and it may be the factor stations in the high latitude region to be affected. As the maximum phase occur on the year 2014, the sunspot activity is highest in this time and may able to influence the H-component of the Earth's magnetic field in contrast to phase occur in year 2011.

Based on the research conducted, a trend can be seen whereas the mean of Earth's magnetic field strength in the Northern Hemisphere increase, the stations in the regions experience a tendency to be antipersistent. Observation on the SAA regions is the low Earth's magnetic field, it exhibit the tendency of the characteristic of the SAA region is to be more persistent and as the mean of the Earth's magnetic field increases, the low latitude region experience an antipersistent characteristics. The trend continue with the increase of the mean of the Earth's magnetic field, stations in the middle latitude regions has more

antipersistent value compare to the low latitude region and SAA regions. As the increase of the mean of the Earth's magnetic field in the high latitude region, it experience a tendency to be antipersistent.

## 4 Conclusions

Based on the analysis, the study of the Sun behaviour during solar cycle 24, in year 2014, the period where the maximum phase of solar cycle 24 occur and in year 2011, the ascending phase of solar cycle 2011 may have an influence on

a region characteristics. Comparing in the solar cycle in year 2011 and year 2014, it is reveal the region tend to be more antipersistent in the ascending phase of solar cycle 24, year 2011 in contrast to the year 2014 where maximum phase of solar cycle 24 happen. Regions such as the high latitude region, middle latitude region and low latitude regions show a more dominant with persistent value during the year 2014. The sun is the main source of ionization in the Earth's upper atmosphere and it may have an effect on the H-component of the Earth's magnetic field to present this type of characteristics. As a

conclusion, compare to other regions, the SAA region has a weak Earth's magnetic field and the characterization of the SAA region has a tendency to be persistent during active period as well as normal period. The SAA region is a region where the number of energetic particles in this region is larger than other regions of space. It is also more vulnerable to radiation from high-energy particles.  The persistent experience means a positive or negative anomaly in the past is more likely to be followed by the same type of anomaly in the future. This may be correlated to a possibility whereby the number of large energetic

particles in the region can be said to remain high in the time to come based on the trend of the persistent value the SAA region exhibit.

## Data Availability

Data for this research can be obtained from INTERMAGNET. It can be accessed at [www.intermagnet.org](www.intermagnet.org).



**Author Contribution**

The research is supervise by Mardina Abdullah and Nurul Shazana Abdul Hamid. Mardina Abdullah contribute in the management of the research, provide authorization as well as approval. Nurul Shazana Abdul Hamid contribute in inventing the coding for the program on the power spectrum analysis and Hurst exponent, give permission to carry on the research and provide response on the research. Khairul Afifi Nasuddin conduct the research based on the supervision of Mardina Abdullah and Nurul Shazana Abdul Hamid and prepared the manuscript with contribution from all authors.

**Competing Interests**

The authors declare that they have no conflict of interest.

**Acknowledgements**

The results presented in this paper rely on data collected at magnetic observatories. We thank the national institutes that support them and INTERMAGNET for promoting high standards of magnetic observatory practice
(www.intermagnet.org).

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
