# Peer review of "South Atlantic Anomaly during ascending and maximum phase of solar cycle 24"

_Nonlinear Processes in Geophysics, 2020_

## Referee Comment (RC1) · Anonymous Referee #1 · 6 Jul 2020

This is a repeat of the analysis of Nasuddin, K. A., Abdullah, M., and Abdul Hamid, N. S.: Characterization of the South Atlantic Anomaly, Nonlin. Processes Geophys., 26, 25–35, https://doi.org/10.5194/npg-26-25-2019, 2019 with all its problems, inconsistencies and poor understanding of the magnetic field, but with a longer time-series.

The authors essentially posit that the long correlation time ("persistant") of the geomagnetic time series recorded within the South Atlantic Anomaly is somehow related to the weaker magnetic field. This is entirely untrue - it is related to the source during quiet time and geomagnetic storms, namely the ring current at low latitudes (not the ionosphere like at high latitudes). This is not mentioned once in the entire manuscript. The ring current changes slowly, is relatively distant ($\sim$6 Re) and has a long response and recovery time. At high latitudes and under the equatorial electrojet (like AAE and TAM)

[Figure]

there are additional proximal current systems which do change rapidly and reduce the correlation time ("antipersistant"). This is the reason - not the main field strength.

How hard would it have been to look at another low latitude sector, like Guam in the Pacific and that area which has a similar spread of geomagnetic ground stations? You would have easily disproved your own point. All observatories at the same latitude experience similar external fields e.g. Cox, G. A., Brown, W. J., Billingham, L., & Holme, R. (2018). MagPySV: A Python package for processing and denoising geomagnetic observatory data. Geochemistry, Geophysics, Geosystems, 19, 3347– 3363. https://doi.org/10.1029/2018GC007714

The work is full of basic errors and shows a real lack of understanding about the Earth's magnetic field. For example, the authors state the SAA is due to the dipole being offset from the axis. This is completely untrue - the SAA is due to the large reversed flux patch on the core-mantle boundary. Look for references from Gubbins from as far back as the 1980s or or Metman et al (2018, PEPI) for examples of what causes the SAA and it's variation.

Please also note the supplement to this comment:
https://npg.copernicus.org/preprints/npg-2020-15/npg-2020-15-RC1-supplement.pdf

**Supplement:**

[revised manuscript text omitted]

---

## Referee Comment (RC2) · Anonymous Referee #2 · 6 Jul 2020

This paper focused on the South Atlantic Anomaly during the ascending and maximum phase of solar cycle 24, the research should be meaningful to reveal the unrevealed mechanisms and features of SAA region. The paper is in good organization, some revisions should be done to improve the quality of paper and get some more solid conclusions for the topic. 1) When the impact of solar activity is considered, it is better to study the entire solar ascending phase, and also make comparison with the feature in solar descending phase. I suggest the authors to make further analysis and comparisons. 2) The geomagnetic storms selected in the present experiment are not very representative in solar cycle 24, severe geomagnetic storms happened in 2015 and 2017 in this solar cycle, the features should be considered and analyzed. 3) The radiation of SAA region during geomagnetic storm can be compared with the normal

condition, if probable, the authors can use some solar radiation flux data to study the solar impacts on SAA during some specific conditions. 4) Some minor language errors should be corrected.

---

## Author Comment (AC1) · 23 Aug 2020

The comment was uploaded in the form of a supplement:
https://npg.copernicus.org/preprints/npg-2020-15/npg-2020-15-AC1-supplement.zip

———————————————

---

## Author Comment (AC2) · 23 Aug 2020

**Response to Referee comments**

**Name of manuscript:** South Atlantic Anomaly during ascending and maximum phase of solar cycle 24

**Authors:** Khairul Afifi Nasuddin, Mardina Abdullah, Nurul Shazana Abdul Hamid

We thank Nonlinear Processes in Geophysics for an experience in improving the journal. The comment has been read and taken consideration discreetly. The following summarize the effort the author take in answering the comment.

**Comments from Referee 2:**

**(1) Comments from Referee 2:**

This paper focused on the South Atlantic Anomaly during the ascending and maximum phase of solar cycle 24, the research should be meaningful to reveal the unrevealed mechanisms and features of SAA region. The paper is in good organization, some revisions should be done to improve the quality of paper and get some more solid conclusions for the topic. 1) When the impact of solar activity is considered, it is better to study the entire solar ascending phase, and also make comparison with the feature in solar descending phase. I suggest the authors to make further analysis and comparisons. 2) The geomagnetic storms selected in the present experiment are not very representative in solar cycle 24, severe geomagnetic storms happened in 2015 and 2017 in this solar cycle, the features should be considered and analyzed. 3) The radiation of SAA region during geomagnetic storm can be compared with the normal condition, if probable, the authors can use some solar radiation flux data to study the solar impacts on SAA during some specific conditions. 4) Some minor language errors should be corrected.

**(2) Author's response:**

The author have studied the comment given by the referee.

1) The author have studied the ascending phase and it reveal SAA region tend to be persistent during year 2011. One of the author research on the ascending phase of solar cycle 24 conduct on 11 March 2011 (active period) and 3 February 2011 (normal period) in journal " Characterization of the South Atlantic Anomaly " reveal SAA region has a tendency to be persistent ". In this journal " South Atlantic Anomaly during ascending and maximum phase of solar cycle 24 ", on 8 August 2011 (active period) and 24 July 2011 (normal period), SAA region also reveal a tendency to be persistent during the ascending phase of solar cycle 24.

2) The author hope to consider on researching on the maximum phase of solar cycle 24. In this period, the maximum phase of solar cycle occur on April 2014.

3) It is a good idea. The author take note regarding the research of SAA region but due to data availability, the component of the Earth's magnetic field is study.

4) The author have inserted a new figure to represent the periodogram previously for figure 2. The new figure concern on the station involve in the research. The author also explain more on the periodogram.